# Safety analysis of China's strategic material maritime Transport Channel based on Bayesian Network

Xue Chen
School of Navigation
Dalian Maritime University
Liaoning, China
E-mail: 1727767908@qq.com

Xin Wang
School of Navigation
Dalian Maritime University
Liaoning, China
E-mail: xin.wang@dlmu.edu.cn

Qing Yu
School of Navigation
Dalian Maritime University
Liaoning, China
E-mail:qing.yu@jmu.edu.cn

Lingling Feng
China Institute of marine
technology & economy
Beijing, China
flingling_1122@126.com

*Abstract*—**Aiming at the influence of the maritime transport channel environment on the safety of maritime transport of strategic materials in China, a safety environment assessment model of the maritime transport channel based on the Bayesian network is proposed. First of all, fully considering the environmental impact factors of traditional maritime transport channels, and combining them with non-traditional factors such as shipping events and sudden public health events, an index system of environmental impact factors of maritime transport channels for strategic materials in China is constructed. Based on expert experience and historical data, a model of environmental factors for marine accident risk at key nodes of maritime transportation channels based on the Bayesian Network (BN) was proposed. At the same time, using the actual data of the key nodes of China's strategic materials maritime transport channel from 2012 to 2022, such as maritime casualties and other related environmental factors, the Expectation Maximization (EM) algorithm is used to compensate for the absence of data and adaptively learn the parameters to update the parameters of the conditional probability table. Finally, the sensitivity analysis of the environmental factors affecting the safety of sea lanes is carried out. It is verified that the BN model can effectively identify the environmental factors of sea transport channels, and can accurately quantify the importance of environmental factors to the safety of sea transport of strategic materials in China. The analysis results show that the channel environment is the main factor affecting the safety of China's strategic materials maritime transport channels.**

*Keywords*—**Strategic materials; Bayesian network; sensitivity analysis; expectation-maximization algorithm**

## I. INTRODUCTION

Under the new situation that strategic and emergency materials have been paid much attention to, maritime transportation has assumed the task of receiving and unloading more than 90 % of China's foreign trade goods, and has received and unloaded about 98 % of the country's foreign trade imports of iron ore, 93 % of foreign trade imports of crude oil, 97 % of foreign trade imports of grain, 89% of foreign trade imports of coal and other strategic materials [1]. The environmental factors of the key nodes of the maritime transport channel will affect the supply chain security of China's strategic materials.

In recent years, domestic and foreign scholars have paid more and more attention to research on the safety risk of strategic materials in maritime transportation. Yu summarized the situation faced by China's grain maritime transportation given the impact of the current Russian-Ukrainian conflict on the global grain transportation pattern [2]. Siddiqui and Verma assessed the risks of intercontinental transport of crude oil by considering specific locations' economic and environmental costs [3]. Zhang X C et al. considered the threat of political risks such as war risks and ethnic conflicts to oil maritime transport [4]. Zeng W G et al. analyzed the current situation of China's coal railway transportation and waterway transportation in recent years through Chart Analysis and put forward the countermeasures to optimize China's coal transportation system [5]. Wang S studied the risk of interruption or blockage of key points in the grain international trade route, and proposed countermeasures [6].

In particular, it should be noted that risk factor identification, risk assessment, and safety assessment are important components of the safety assessment of maritime transport channels. For this reason, many scholars have further studied the key nodes of maritime transport channels in relevant aspects. Gao et al. proposed a method to identify the risk factors of sea lanes from the perspective of the coupling effect, aiming at the characteristics of many interrelated risk factors of sea lanes [7]. Li Jing et al. established an evaluation index system based on the characteristics of the factors that threaten the safety of sea lanes and established a factor-two-factor evaluation model to comprehensively evaluate the safety status of China's sea lanes [8]. Jia S used the fuzzy analytic hierarchy process to evaluate the security situation of the international channel in the South China Sea [9]. Li et al. established a blind number model to evaluate the safety of China's maritime strategic channels through expert scoring [10]. Jiang M proposed a new strait/canal security assessment framework based on a fuzzy evidence reasoning method to assess the security of MSR coastal straits/canals, taking into account the impact of various non-traditional security factors such as pirate attacks, terrorism, and accidents [11]. Liu used the generalized analytic network process (G-ANP) to conduct a comprehensive analysis of the risks of key nodes in China's sea lanes [12].

In addition, Bayesian networks have been widely used in risk analysis and sensitivity analysis in various fields. Liu Jia et al. used fault tree analysis and the Bayesian network model to

This work is supported in part by the National Natural Science Foundation of China (Grant. 51909022).

analyze the vulnerability of the port coal supply chain under the influence of capacity bottlenecks and seasonal cycles [13]. Ding-Hao et al. used a fault tree and Bayesian network to construct a brittleness risk assessment model and analyzed the important reasons for the brittleness risk in the maritime transportation of imported Middle East oil [14]. Based on Bayesian theory, Lv Jing et al. constructed an analysis model of risk factors for sea lanes and conducted a sensitivity analysis on risk factors for sea lanes [15]. Jiang M integrated the EM algorithm and historical cases to construct a Bayesian network model for identifying the risk factors of key nodes in maritime transport channels [16].

Based on the existing research, aiming at the environmental conditions of maritime casualties at the key nodes of maritime transport channels, this paper further considers the sources of risk factors, integrates traditional security and non-traditional security threat factors, and establishes an index system of environmental impact factors of maritime transport channels for strategic materials in China. Because the data of various environmental factors are qualitative and quantitative, incomplete and uncertain, the BN theory can better make up for the relevant defects. At the same time, BN has good forward-backward reasoning ability, which can more accurately capture and analyze the causal relationship between factors. Combined with the EM algorithm, it can effectively deal with the missing values in the data set and avoid the information loss and analysis deviation caused by the missing data. Therefore, this paper uses the BN model to study the environmental safety of the key nodes of China's strategic materials maritime transport channel and analyzes the probability dependence between the influencing factors.

## II. ESTABLISHMENT OF THE INDEX SYSTEM OF ENVIRONMENTAL IMPACT FACTORS OF CHINA'S STRATEGIC MATERIALS MARITIME TRANSPORT CHANNEL

Maritime transport channels are faced with many complex environmental impact factors. This paper systematically analyzes the risk events of key nodes in maritime transport channels in history and comprehensively considers the environmental impact factors of key nodes in maritime transport channels. The index system of environmental impact factors of China's strategic materials maritime transport channels was established.

### A. Analysis of influencing factors of maritime transport safety

The RIF (Risk Influence Factors) needs to be considered to determine the developed BN model, which has an important comprehensive impact on the reliability of the model. This study combines a literature review, existing databases, and variables from previous studies to identify RIF. Luo and Shin et al. [17] reviewed the research related to maritime accidents and provided important evidence of natural environmental factors such as wind, wave, and visual conditions in the analysis of maritime accidents. Burmeister and others believe that low visibility and bad weather conditions will significantly affect the navigation of ships [18]. Wu et al. proposed three main natural variables, including wind, wave, and visibility, to assess the risk of ship collision [19]. Zhang mentioned that LNG ships have very strict requirements for the channel. The navigation elements of the channel, such as the width of the channel (or

effective width), the depth of the channel, and the width of the curved section of the channel, are the key factors for the navigation safety of LNG ships [20]. Wang took the navigation density (ship density in the navigation waters) as the risk factor of the coupling index system of homogeneous factors of marine ship accidents in the study of the risk mechanism of marine ship accidents [21]. Gong and Lu et al. fully considered the substitutability of waterways in the study of risk assessment of the Maritime Silk Road [22]. Yan and Lam considered political risk factors such as political stability when studying important transport channels such as the Malacca Strait and the Maritime Silk Road [23,24]. Zhang Xuan Cheng et al. also took into account the threat of political risks such as war risks and ethnic conflicts to oil maritime transportation [4]. In the study of China's maritime transport channel security, Ma [25] took the number of countries and military bases in the maritime channel as the variables of the dynamic evolution model of maritime transport channel security vulnerability. Mokhtari studied the role of relevant laws and regulations in port and channel risk management [26]. Zhang and Yang considered the management of the straits and canals, such as the standardization of security and the number of organizations, in the risk assessment of maritime transport channels [27,28]. Jiang M Z regards piracy and terrorist attacks as non-traditional security environment factors and combines traditional security factors to conduct risk assessment and early warning research on key nodes of maritime transport channels [16]. Zhu mentioned that emergencies such as the outbreak of COVID-19 and the shutdown of the Suez Canal would lead to the interruption of key nodes of grain maritime transportation, which had a great impact on the timeliness and safety of grain transportation [29]. According to the analysis of historical cases, considering that shipping events and sudden public health events will also affect the normal navigation of ships, they are included in the event condition environment. TABLE I summarizes the environmental impact factors of the risk of key nodes in maritime transport channels.

TABLE I. ENVIRONMENTAL IMPACT FACTORS OF RISK AT KEY NODES OF MARITIME TRANSPORT CHANNEL

| Main influencing factors | Secondary influencing factors | Factor description | source |
|---|---|---|---|
| Natural environment | Extreme severe weather | The presence of extreme severe weather during the Strait / Canal accident | [17] [18] [19] |
| | Visibility | Visibility conditions in the event of the Strait / Canal accident | |
| | Wind | Wind speed in case of the Strait / Canal accident | |
| | Wave | Wave height situation at the time of the Strait / Canal accident | |
| Channel environment | Alternative options | The Strait / Canal can be replaced by cases | [20] [21] [22] |
| | Width | The Strait / Canal width | |
| | Depth | The Strait / Canal water depth | |
| | Ship density | Ship density in the Strait / Canal vessel unit area | |
| | War risk | The Strait / Canal is the risk situation of a war | [4] [23] [24] |

| | | | |
|---|---|---|---|
| Military political environment | Number of affiliated countries | Number of countries to which Straits/Canals belong | |
| | Military base | The presence of military bases at the Strait / Canal | |
| Law and the international environment | Safeguard standardization | The number of specialized governing agencies for the safety of the Strait / Canal country | [26] [27] [28] |
| | Domestic and foreign regulatory legal constraints | Number of relevant legislation and policies in the Strait / Canal host country or internationally | |
| | Manage the number of guarantee organizations. | Number of relevant organizations coordinated with the Strait / Canal host country or internationally | |
| Event condition environment | Piracy | Whether there were pirate attacks at the Strait / Canal | [16] [29] |
| | Terrorist attack | Was there a terrorist attack at the Strait / Canal | |
| | Shipping events | Shipping events at the Strait / Canal | |
| | Sudden public health events | Emergency public health events at the Strait / Canal | |

## B. Index system of environmental impact factors of China's strategic materials maritime transport channel

Based on the identification results of the environmental impact factors of the risks of the key nodes of the maritime transport channel mentioned above, and by the principles of selection of basic indicators such as scientific, conciseness, representativeness, etc., and concerning the selection of factors influencing the maritime security accidents in previous literature, the indicator system of the environmental impact factors of the maritime transport channel of strategic materials in China is constructed. The index system fully considers the diversity and timeliness of event conditions and adds two indicators of shipping events and sudden public health events, which can be used as a general framework for risk research on key nodes of multiple strategic materials maritime transport channels, as shown in TABLE II.

TABLE II. INDEX SYSTEM OF ENVIRONMENTAL IMPACT FACTORS OF CHINA'S MARITIME TRANSPORTATION CHANNEL FOR STRATEGIC MATERIALS

| Results | First-level indicators | Second-level indicators |
|---|---|---|
| Channel risk | Natural environment | Extreme severe weather |
| | | Visibility |
| | | Wind |
| | | Wave |
| | Channel environment | Alternative options |
| | | Width |
| | | Depth |
| | | Ship density |
| | Military political environment | War risk |
| | | Number of affiliated |
| | | Military base |
| | Law and the international environment | Safeguard standardization |
| | | Domestic and foreign regulatory legal constraints |

Manage the number of

| | |
|---|---|
| Event condition environment | Piracy |
| | Terrorist attack |
| | Shipping events |
| | Sudden public health events |

## III. MODELLING AND DATA PREPARATION OF ENVIRONMENTAL FACTORS AT KEY NODES OF MARITIME TRANSPORT CHANNELS

### A. Bayesian networks and their applications

Bayesian network is a graphical reasoning technology, that can integrate different types of subjective and objective information in the presence of uncertainty, different standards, and units. For readers who do not have relevant background knowledge, the graphical representation of variable combinations can help them understand the probability distribution of variable combinations and the relationship between different variables.

In addition, the BN model is widely used in maritime accident analysis and piracy event analysis. In previous studies, the construction of Bayesian networks mainly depends on two methods: data-driven and expert knowledge-driven. The data-driven method learns the dependencies between variables from a large amount of data, but successful implementation not only depends on large-scale, high-quality data but also requires reasonable training strategies and strict verification of model reliability. The expert knowledge-driven method relies on the experience of domain experts to construct the Bayesian network structure, which may be affected by the subjective factors of experts. Therefore, this paper combines expert experience and data-driven methods to construct a BN model of environmental risk factors for key nodes of maritime transport channels.

BN is defined as a unary cubic group $G=（N，S，C）$ on the variable set $X=\{X_1，X_2，…，X_n\}$, where $N$ is the set of nodes, $S$ is the set of directed edges, and $C$ represents the conditional probability distribution between network nodes. Each node represents an event as a random variable in the model, and its joint probability distribution is calculated as follows (1):

$$P\left(X_1, X_2, \cdots X_n\right) = \prod_{i=1}^{n} P\left(X_i \mid P_a\left(X_i\right)\right) \qquad (1)$$

$P_a(X_i)$ is the parent node of $X$, and if there is no parent node, it is the root node. $P(X_i \mid P_a(X_i))$ denotes the conditional probability distribution of $X_i$.

The BN construction process includes two parts: model topology identification and model parameter training.

### B. Model topology identification

According to the analysis of the environmental impact factors of the key nodes of China's maritime transport channels for strategic materials, the database and survey results of Marine Casualties and Incidents and Piracy and Armed Robbery in the Global Integrated Shipping Information System (GISIS) of the International Maritime Organization (IMO) were collected, and the current research literature of experts and scholars on maritime transport channels was comprehensively considered. Based on the index system of environmental impact factors of

China's strategic materials maritime transport channel, this paper constructs the BN topology, as shown in Fig. 1. The network consists of 24 leaf nodes and several directed line segments. The root nodes are Wind, Wave, Visibility, Extremely severe weather, Width, Depth, Alternative options, Ship density, War risk, Number of subordinate countries, Military base, Safeguard standardization, Domestic and foreign regulatory legal constraints, etc. The intermediate nodes are the Natural environment, Channel environment, Military and political environment, Law and international environment, and Event condition environment. The leaf node is the Channel risk.

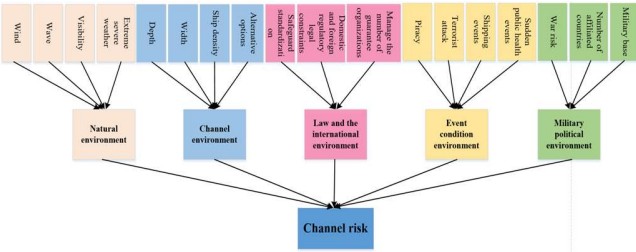

Fig. 1. Bayesian network topology

*C. Model parameter training*

Since the subjectivity of expert knowledge affects the accuracy of network parameters, this paper combines expert knowledge and machine learning techniques to achieve parameter training of BN models using data-driven. The acquisition of parameter training samples includes two parts: data collection and data processing.

- Data collection.

According to the distribution of China's maritime import and export of strategic materials in recent years, important straits and canals through which strategic materials transport channels pass are screened. From the GISIS database [21], 364 maritime casualty accidents in important straits and canals from 2012 to 2022 were screened and downloaded, of which 340 occurred on strategic materials transport ships. Except for 156 accidents caused by human factors and ship factors, the remaining 184 accidents were caused by environmental factors, accounting for 54 %. A total of 2635 pirate terrorist attacks in global waters from 2012 to 2022 were screened and downloaded, of which 999 occurred in important straits and canals through which strategic material transport channels passed, accounting for up to 37.9 %. And 814 cases occurred on strategic materials transport ships, accounting for up to 81.5 %. In summary, this paper takes 184 maritime casualties caused by environmental factors and 814 pirate terrorist attacks affected by the environmental impact of emergencies as the research objects and analyzes the environmental factors of important shipping channels of strategic materials. Taking a historical case as sample data, each data contains data from 18 root nodes, including wind, wave, visibility, extreme weather, width, depth, substitutability, etc., all of which constitute the sample data training set. The data acquisition channels of each environmental impact factor are shown in TABLE III.

TABLE III. DATA SOURCES OF ENVIRONMENTAL FACTORS AT KEY NODES OF MARITIME TRANSPORT CHANNELS

| Environmental category | risk factor | data sources |
|---|---|---|
| Channel environment | Alternative options | Qualitative evaluation of relevant data |
| | Width | https://www.shipxy.com, Hifleet Online (www.hifleet.com) |
| | Depth | https://www.shipxy.com, Hifleet Online(www.hifleet.com) |
| | Ship density | https://www.marinetraffic.com/en/ais/home/centerx:32.7/centery:31.9/zoom:9,HifleetOnline |
| Natural environment | Extreme severe weather | https://www.ncei.noaa.gov/access/search/data-search/global-summary-of-the-day? |
| | Visibility | https://www.ncei.noaa.gov/access/search/dataset-search?observationTypes=Ocean |
| | Wind | National Center for Environmental Information (NCEI) (NOAA. gov) |
| | Wave | National Center for Environmental Information (NCEI) (NOAA. gov) |
| Law and the international environment | Safeguard standardizatio n | National laws and regulations database (npc.gov.cn) |
| | Domestic and foreign regulatory legal constraints | International Country Risk Guide (https://epub.prsgroup.com/products/icrg/), National Laws and Regulations database (npc.gov.cn) |
| | Manage the number of guarantee organizations. | Qualitative evaluation of relevant data, https://www.imo.org/en/About/Pages/DocumentsResources.aspx |
| Event condition environment | Piracy | GISIS - Piracy and Armed Robbery (https://gisis.imo.org/Public/Default.aspx) |
| | Terrorist attack | Global Terrorism Database (GTD) (https://gtd.terrorismdata.com/), GISIS - Piracy and Armed Robbery (https://gisis.imo.org/Public/Default.aspx) |
| | Shipping events | https://www.wri.org/data |
| | Sudden public health events | https://www.who.int/zh/emergencies/diseases/novel-coronavirus-2019 |
| Military political environment | War risk | https://www.lmalloyds.com/LMA/Underwriting/Marine/Joint_War_Committee/ |
| | Number of affiliated countries | Open Street Map (https://www.openhistoricalmap.org/) |
| | Military base | Military Times, U.S. State Department, Global Security (https://www.globalsecurity.org/military/),(https://www.ntsb.gov/safety/safety-studies/Pages/SafetyStudies.aspx) |

- Data processing.

Since the BN model parameter training algorithm requires the node data to be discrete values, it is necessary to discretize the sample data. The detailed root node state settings as shown in TABLE IV are defined and explained by drawing on references in related fields or relevant information on key nodes, and the individual variables are discretized by referring to the existing risk research experience in related fields, and TABLE V shows the basis and details of the hierarchical classification of the root node variables.

TABLE IV. CLASSIFICATION DESCRIPTION OF BN NODE VARIABLES

| Variable | States |
|---|---|
| Wind | s1, s2, s3= {low, medium, high} |
| Wave | s1, s2, s3= {low, medium, high} |
| Visibility | s1, s2, s3= {good, medium, poor} |
| Extreme severe weather | s1, s2= {no, yes} |
| Depth | s1, s2, s3= {deep, medium, shallow} |
| Width | s1, s2, s3= {broad, medium, narrow} |
| Ship density | s1, s2, s3= {less, normal, more} |
| Alternative options | s1, s2, s3= {more, normal, less} |
| Safeguard standardization | s1, s2, s3= {more, normal, less} |
| Domestic and foreign regulatory legal constraints | s1, s2, s3= {more, normal, less} |
| Manage the number of guarantee organizations | nods1, s2, s3= {more, normal, less} |
| Piracy | s1, s2= {no, yes} |
| Terrorist attack | s1, s2= {no, yes} |
| Shipping events | s1, s2, s3= {no, less, more} |
| Sudden public health events | s1, s2, s3= {less, normal, more} |
| War risk | s1, s2, s3= {low, medium, high} |
| Number of affiliated countries | s1, s2, s3= {less, normal, more} |
| Military base | s1, s2= {no, yes} |
| Natural environment | s1, s2, s3= {good, medium, poor} |
| Channel environment | s1, s2, s3= {good, medium, poor} |
| Law and the international environment | s1, s2, s3= {good, medium, poor} |
| Event condition environment | s1, s2, s3= {good, medium, poor} |
| Military political environment | s1, s2, s3= {good, medium, poor} |
| Channel risk | s1, s2, s3= {low, medium, high} |

TABLE V. THE HIERARCHICAL CLASSIFICATION OF THE BN NODE VARIABLES

| Variable | Variable description | s1 | s2 | s3 |
|---|---|---|---|---|
| Wind | Beaufort wind scale | [0,4] | [5,8] | [9,12] |
| Wave | Sea condition level | [0,3] | [4,6] | [7,9] |
| Visibility | Sea surface visibility | [7,9] | [4,6] | [0,3] |
| Extreme severe weather | Meteorological events | not have | have | |
| Depth | D (m) | [30, +∞] | [10,30) | (0,10) |
| Width | W (n mile) | [20, +∞] | [1,20) | (0,1) |
| Ship density | S (0.08 km² / year) | (0,221) | [221,50k) | [50k, +∞] |
| Alternative options | Number of routes | [5, +∞] | [2,5) | [0, 2) |
| Safeguard standardization | The number of governing bodies | [5, +∞] | [3,5) | [0, 3) |
| Piracy | | not have | have | |
| Terrorist attack | | not have | have | |
| Shipping events | frequency | 0 | 1 | 2 or more |
| Sudden public health events | frequency | 1 | 2~3 | 4~5 |
| War risk | War area | not have | far | near |
| Number of affiliated | | 1 | 2 | 3 |
| Military base | | not have | have | |
| Channel risk | Accident severity | Less serious | Serious | Very serious |

## IV. EXAMPLE ANALYSIS

### A. Model training based on the EM algorithm

Due to the incomplete data collected and limited sample size, the EM algorithm was used to train the model parameters on the training samples with missing data to maximize the learning and modeling of the relationship between the influencing factors and the level of risk.

- EM algorithm

EM algorithm is an iterative algorithm for parameter estimation, especially in the case of missing data. The basic idea of the EM algorithm is: firstly, according to the existing observation data, the value of the model parameters is estimated; then, the value of the missing data is estimated according to the model parameter value, and then the parameter value is estimated again according to the missing data value and the previously observed data, and then iterate repeatedly to find the maximum likelihood estimation of the parameter $\theta$. Each iteration of the EM algorithm includes two main steps: the expectation step (E-step) and the maximization step (M-step). The incomplete data set $Z = (X, Y)$ is defined. $X$ is the complete part of the data set, and Y is the incomplete part of the data set.

Expectation step (E-step): calculate the expectation of the log-likelihood function for the incomplete data set $Z = (X, Y)$ given the observed data $X$ based on the current parameter values $\theta$. For this purpose, define the expectation of the log-likelihood function as:

$$Q(\theta \mid \theta^{(k)}) = E\{logf(Z) \mid \theta^{(k)}\} \qquad (2)$$

Where $f(Z)$ is defined as the likelihood function of the incomplete data set $Z$ given $X$ and $\theta$. The parameter $\theta^{(k)}$ is the observed value $X$ of the given complete part and the parameter value obtained by the last k iterations.

Maximization step ( M-step ): In this step, the algorithm attempts to maximize the expectation of the log-likelihood function under complete data, that is, selecting a parameter $\theta^{(k+1)}$ to satisfy the following formula :

$$\theta^{(k+1)} = arg\, \underset{\theta}{m}ax\, Q(\theta \mid \theta^{(k)}) \qquad (3)$$

- Training results

The conditional probability table (CPT) of the intermediate and final nodes was established using the IF-THEN method, which helps to prevent the model from overfitting the noise in

the data, and the preprocessed data set was input into the NETICA software, and the EM algorithm shown in (2) and (3) was used to learn from the sample data and deal with the missing data of the samples, and the CPT was adjusted after the parameter training was conducted. Probability values in the CPT to reduce the bias and improve the accuracy of the prediction.

Select 30% of the sample data as a test set to verify the prediction results of the samples, the ratio of the prediction error to the total data is regarded as the error rate index, and the results show that the error rate of the prediction of 30% of the samples is 16.33%, and the overall sample prediction result is 18.37%, which shows that the framework is reasonable and the prediction effect is good. At the same time, the algorithm is used to update and regulate the conditional probability of each node, and the training results are shown in Fig. 2. From Fig. 2, it can be seen that the a priori probability of the root node, for example, the depth of the channel, is at deeper depths with a probability of 43.0%, at intermediate depths with a probability of 35.7%, and shallower depths with a probability of 21.3%. When the a priori probability of an intermediate node variable changes, the probability of channel risk changes accordingly. The model can be used to analyze the degree of influence of a node on channel risk, and the probabilistic dependence between the query node 'channel risk' and its parent node is shown in Fig. 3. Favorable environmental factors can increase the probability of lower channel risk.

To better optimize the output of the model, it is necessary to process the distributed probability output to obtain a specific risk value. In this paper, we use the utility function in Fig. 4 to process the distributed results of the risk probability to obtain a comprehensive risk value, which can provide a more intuitive basis for decision-making for maritime stakeholders.

Cannot directly observe the conditional probability distribution between the node and the parent node from Fig. 2, due to more nodes in this paper, and the state of each node is also 2~3, the whole model has a larger collection of CPT data, so this paper lists some of the CPT, see TABLE VI.

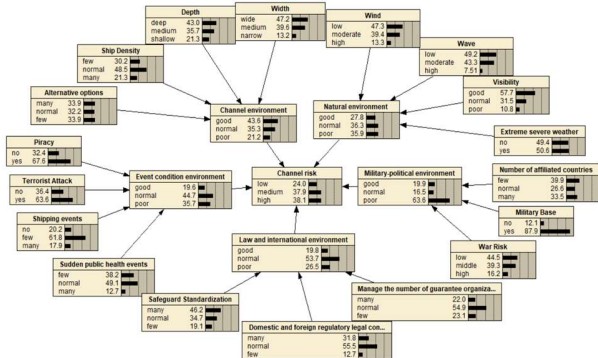

Fig. 2. BN model training results

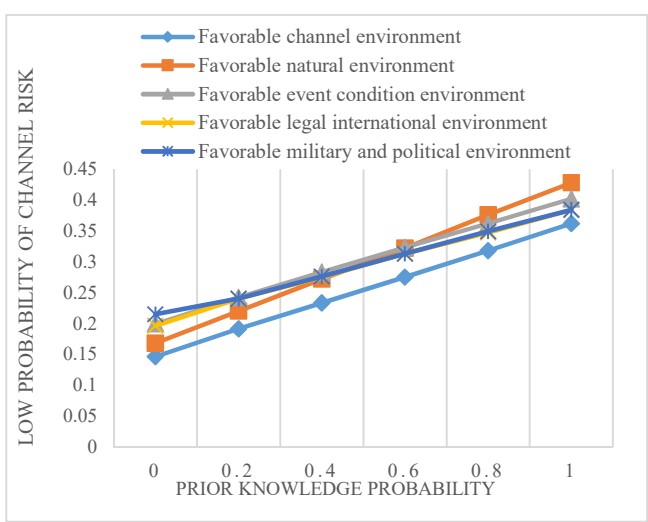

Fig. 3. Low-risk probability distribution of waterways based on prior knowledge

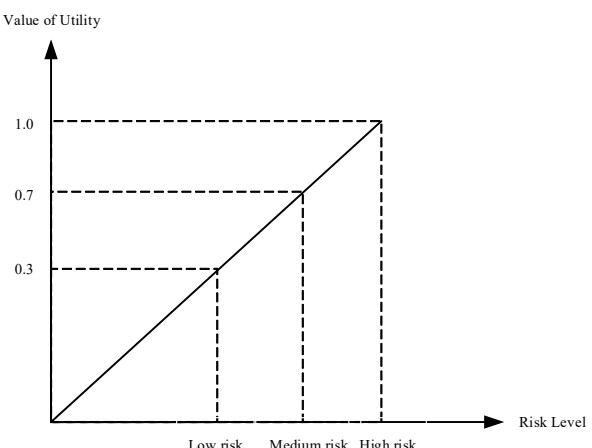

Fig. 4. Utility function

TABLE VI. DEMONSTRATION OF CPT AT KEY NODE RISKS

| Parent node | | | | | Sub-nodes (critical channel risk) | | |
|---|---|---|---|---|---|---|---|
| Channel environment | Natural environment | Event condition environment | Law and the international environment | Military political environment | Low risk | Medium risk | High risk |
| s1 | s1 | s1 | s1 | s1 | 99.99 | 0.002 | 0.003 |
| s1 | s1 | s1 | s1 | s2 | 79.99 | 19.999 | 0.003 |
| s1 | s1 | s1 | s1 | s3 | 80.02 | 0.002 | 19.972 |
| s1 | s1 | s1 | s2 | s1 | 79.99 | 19.999 | 0.002 |
| s1 | s1 | s1 | s2 | s2 | 61.96 | 38.030 | 0.002 |
| s1 | s1 | s1 | s2 | s3 | 60.04 | 19.983 | 19.968 |

| | | | | | | | |
|---|---|---|---|---|---|---|---|
| s1 | s1 | s1 | s3 | s1 | 79.81 | 0.002 | 20.183 |
| s1 | s1 | s1 | s3 | s2 | 59.99 | 20.002 | 20.002 |
| s1 | s1 | s1 | s3 | s3 | 57.81 | 0.002 | 42.183 |

## B. Sensitivity analysis and results

- Sensitivity analysis

The sensitivity analysis of BN is to analyze the probability change of the query node (that is, the degree of reduction of the information entropy of the query node) by changing the probability estimation of the network evidence node, to judge the evidence node that has the greatest impact on the query node. The evidence node in this paper refers to the observable root node, such as Wind, Wave, etc. The query node refers to the Channel risk, and the intermediate node refers to the Natural environment, Channel environment, etc. Information entropy is a statistic that describes the degree of dispersion of random variables. The greater the information entropy, the higher the degree of uncertainty of variables. The calculation formula (4) is as follows:

$$H(Y) = -\sum_{y \in Y} P(y) \log P(y) \tag{4}$$

Mutual information represents the amount of information shared between two variables. It is a measure of the degree of interdependence of variables and is used to calculate the strength of the relationship between query nodes and evidence nodes. The degree of information entropy reduction of query nodes is calculated under different states of evidence nodes. The mutual information of two discrete random variables $X$ and $Y$ can be defined as:

$$H(X;Y) = \sum_{y \in Y} \sum_{x \in X} P(x,y) \log \left( \frac{P(x,y)}{p(x)p(y)} \right) \tag{5}$$

Equation (5) where $P(x, y)$ denotes the joint probability distribution function of X and Y, and $p(x)$ and $p(y)$ denotes the marginal probability distribution function of $X$ and $Y$, respectively. Based on the mutual information to find out the influencing factors that have the greatest dependence on the query node, and using the results of the training of the Bayesian network model parameters, the query node is subjected to inference analysis:

$$p(Q \mid E) = \frac{p(Q,E)}{p(E)} \tag{6}$$

In (6), $Q$ and $E$ represent the query node and evidence node respectively. Since the Bayesian network sensitivity analysis reasoning is an NP-hard problem (Non-deterministic Polynomial-time hard problems), this paper uses the joint tree algorithm to reason the sensitivity analysis of the model.

- Results of sensitivity analysis

Using the software NETICA, the mutual information between the channel risk node and its parent node is shown in TABLE VII. It can be seen from TABLE VII that the channel environment, natural environment, event condition environment, Law and the international environment, military and political environment are strongly correlated with the risk of key nodes. According to the training results of model parameters, the

inference algorithm of the joint tree algorithm is used to analyze the sensitivity of nodes with strong correlation and the influence degree of this influencing factor on the risk of key nodes is discussed. The results are shown in TABLE VIII.

TABLE VII. MUTUAL INFORMATION BETWEEN THE WATERWAY RISK NODE AND THE PARENT NODE

| Nodal variables | Mutual information | Percentage % |
|---|---|---|
| Channel risk | 1.55496 | 100 |
| Channel environment | 0.07650 | 4.920 |
| Natural environment | 0.06329 | 4.070 |
| Event condition environment | 0.04440 | 2.860 |
| Law and the international environment | 0.04432 | 2.850 |
| Military political environment | 0.04220 | 2.710 |
| Manage the number of guarantee organizations. | 0.01179 | 0.758 |
| War risk | 0.01138 | 0.732 |
| Number of affiliated | 0.01062 | 0.683 |
| Wind | 0.00787 | 0.506 |
| Domestic and foreign regulatory legal constraints | 0.00784 | 0.504 |
| Shipping events | 0.00604 | 0.388 |
| Extreme severe weather | 0.00398 | 0.256 |
| Ship density | 0.00290 | 0.186 |
| Military base | 0.00282 | 0.182 |
| Sudden public health events | 0.00273 | 0.176 |
| Depth | 0.00214 | 0.138 |
| Safeguard standardization | 0.00200 | 0.128 |
| Piracy | 0.00193 | 0.124 |
| Alternative options | 0.00168 | 0.108 |
| Width | 0.00154 | 0.099 |
| Terrorist attack | 0.00152 | 0.098 |
| Wave | 0.00108 | 0.070 |
| Visibility | 0.00053 | 0.034 |

TABLE VIII. SENSITIVITY ANALYSIS OF "WATERWAY RISK"

| Key Node Risks | | High risk probability value (%) | | |
|---|---|---|---|---|
| BN probability value (%) | Nodal variables—good | Take 0 when the value | Take 100 when the value 100 | Difference range |
| 43.6 | Channel environment—good | 49.0 | 40.9 | 8.1 |
| 27.8 | Natural environment—good | 36.3 | 18.8 | 17.5 |
| 19.6 | Event condition environment—good | 23.9 | 15.9 | 8.0 |
| 19.8 | Law and the international environment—good | 24.0 | 15.9 | 8.1 |

| 19.9 | Military political environment —good | 25.7 | 12.7 | 13.0 |

Sensitivity analysis is an effective method to evaluate the main factors that play a major role in the risk of key nodes. It can be seen from TABLE VIII that the most sensitive evidence node is the 'Natural environment '. Changes in the probability of this node will lead to a high-risk probability of key nodes as low as 18.8 % or as high as 36.3 %. However, on the whole, 27.8 % of the risk events occur in good conditions, and only 10.8 % of the visibility is low. The wind and waves of the natural environment of most risk events are in normal or good condition. Therefore, the natural environment usually provides good conditions (64.1%).

The change in the probability of a favorable channel environment will lead to a fluctuation of 8.1 % of the high-risk probability of key nodes, as low as 40.9 % or as high as 49.0 %. The impact of the parent node ' shipping event ' of the event condition environment on the safety of key nodes is mainly reflected in the fact that when a maritime accident occurs in a narrow channel, it will lead to the blockage of the channel and the paralysis of transportation. When the environment of other events is in a favorable situation, the occurrence of shipping events will increase the high-risk probability of channel risk to 19.4 %.

The law and international environment perform relatively well (73.5%) despite the presence of strong sensitivities, with most of the critical nodes now being managed by specialized management structures, and various policies and laws being introduced internationally to secure the critical nodes.

In addition, the military-political environment and event conditions environment also have a greater impact on the risk probability of critical nodes, the main factors of which are that 33.5% of the events occur in the channel area belongs to more than one country, and 55.5% of the events have the impact of war risk. The frequent occurrence of military and political activities in the coastal countries of the channel will reduce the loading and unloading efficiency and passing capacity of the key nodes on the channel, and even block the navigation of ships, thus causing the blockage of the sea transportation channel.

From TABLE VII, it can be seen that the mutual information between the channel environment and the risk of key nodes is the largest, implying that the channel environment is the most important factor affecting the risk of key nodes. Among the mutual information of all root nodes and the five intermediate nodes mentioned above, wind, ship density, shipping events, the number of management and security organizations, and the number of countries affiliated with them are the root nodes that have the greatest influence on the fairway environment, the natural environment, the environment of the event conditions, the law and international environment, and the military-political environment, respectively, and the ratio of mutual information reaches 53.57%, 44.00%, 48.23%, 37.74%, and 55.98%.

## V. CONCLUSIONS AND DISCUSSIONS

After describing the problems to be studied, this paper analyzes and selects the key nodes of the maritime transport channel, and clarifies the research object of this paper. Through the statistical analysis of the relevant literature and the environmental impact factors of maritime casualties in the past 10 years, the environmental impact factors of maritime accidents are analyzed from five aspects: channel environment, natural environment, event condition environment, law and international environment, and military and political environment. The traditional security threats and non-traditional security threats are fully considered, and the existing environmental impact factor identification model is improved. Make the model more in line with the current background of China's strategic materials maritime transport channel key nodes of the security risk situation research. At the same time, the proposed BN model is constructed based on previous excellent papers, and its rationality and effectiveness are proved. Based on the actual data of maritime casualties and piracy terrorist attacks in major straits and canals from 2012 to 2022, the EM algorithm is used to train the parameters of the Bayesian network analysis model. According to the training results of model parameters, the sensitivity analysis of the main influencing factors of the environment of maritime casualties is carried out by using the joint tree reasoning algorithm. The influence degree of channel environment, natural environment, event condition environment, law and international environment, and military-political environment are discussed respectively, which provides a reference for the follow-up research of risk response measures.

Since the maritime casualty event is a complex event involving many natural, human, and social factors, there are still many influencing factors to be considered, such as ship factors and human factors. However, due to its strong subjectivity, it is difficult to obtain and quantify. Therefore, future research focuses on how to introduce more influencing factors and establish a more reasonable and comprehensive marine casualty accident assessment model.

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
