# OpenReview forum: "Safety analysis of China's strategic material maritime Transport Channel based on Bayesian Network"
_IEEE.org/ICIST/2024/Conference — IEEE ICIST 2024 Conference Submission_

### Official Review · Reviewer_GLXT · 2024-08-25
**Accept, some mistakes should be revised**

**Rating:** 7
**Confidence:** 3

**Review:**

1. The typeface of some variables are improper, such as “k” below formula (2).
2. Table VII contains a lot of information with insufficient descriptions in the paper.
3. What is the unit of mutual information in Table VII. Can it be quantified?

---

### Official Review · Reviewer_Zc3V · 2024-09-03
**Safety analysis of China's strategic material maritime Transport Channel based on Bayesian Network**

**Rating:** 7
**Confidence:** 4

**Review:**

1.The authors have presented a well-structured and comprehensive framework for assessing the safety of China's strategic material maritime transport channel using Bayesian Network (BN) models. The integration of traditional and non-traditional risk factors such as shipping events and sudden public health events is innovative and provides valuable insights into the complex environmental factors affecting the safety of strategic material transport. The use of the Expectation Maximization (EM) algorithm to address data gaps is particularly commendable and enhances the robustness of the proposed model. Overall, the study offers a unique perspective and an effective tool for safety analysis in this crucial domain.

2.The authors have conducted a detailed analysis of environmental factors using actual data spanning a decade (2012-2022), which demonstrates a rigorous and practical approach. The sensitivity analysis performed to identify the most significant factors affecting channel safety is insightful and has important practical implications for policymakers and industry stakeholders. The inclusion of shipping events and sudden public health events as environmental factors expands the scope of traditional risk assessment and highlights the need for a holistic view of safety risks. The results, if implemented, could significantly improve the safety and efficiency of strategic material maritime transport in China. However, further discussion on the specific policy recommendations and operational strategies based on the findings would strengthen the paper's practical significance.

---

### Decision · Program_Chairs · 2024-09-06

Accept (Oral)